# Long-Chain Modification of the Tips and Inner Walls of MWCNTs and Their Nanocomposite Reverse Osmosis Membranes

**DOI:** 10.3390/membranes12080794

**Published:** 2022-08-18

**Authors:** Qing Li, Dengfeng Yang, Qingzhi Liu, Jianhua Wang, Zhun Ma, Dongmei Xu, Jun Gao

**Affiliations:** 1College of Chemical and Biological Engineering, Shandong University of Science and Technology, Qingdao 266590, China; 2College of Chemistry and Pharmaceutical Science, Qingdao Agriculture University, Qingdao 266109, China

**Keywords:** multi-walled carbon nanotubes, tip/inner walled modification, polyamide, nanocomposite membranes, long aliphatic chain, water flux, desalination

## Abstract

Multi-walled carbon nanotubes (MWCNTs) were modified on the tips and inner walls by 12-chloro-12-oxododecanedioic acid-methyl ester groups and then added to the polyamide composite membranes to prepare MWCNT-CH_2_OCOC_12_H_23_O_2_ membranes for desalination. The characterization results of transmission electron microscopy, Fourier transform, infrared transform, and thermogravimetric analysis showed that the 12-chloro-12-oxododecanedioic acid-methyl ester group was successfully grafted to the entrances and inner walls of the MWCNTs. The performance of the MWCNTs’ composite membranes was evaluated by scanning electron microscopy, contact angle, and filtration test. The modified membrane morphology is more uniform, and there is no structural damage. The grafting of carbon nanotubes with methyl 12-chloro-12-oxydodecyldicarboxylate could improve the hydrophilicity of the membrane. Under identical conditions, the water flux of MWCNT-CH_2_OCOC_12_H_23_O_2_ membranes was higher than that of the pristine carbon nanotube’s membrane, and the desalination rate was also slightly improved.

## 1. Introduction

With the rapid development of society and the economy, as well as the pollution of water resources along with the increase in population, the global water shortage has been a significant problem in the world. In order to alleviate the shortage of water resources, more efforts have been devoted to design a reverse osmosis membrane with high water flux and salt rejection to desalinate sea water and brackish water [1,2]. Recently, the polyamide (PA) membrane was widely used, due to its good mechanical, endurable, and chemical resistant characteristics and self-lubrication, as well as excellent membrane-separation performance. However, the polyamide membrane still has some problems, such as low permeate flux and biofouling [3].

Carbon nanotubes (CNTs) were discovered by Iijima [4] in 1991, and have many interesting properties, including excellent mechanical [5], electrical [6], thermal properties [7], and so on, which have been extensively used in chemistry, physics, engineering, and other fields, such as nano-probes, sensors, drug delivery, cell, energy storage, etc. [8,9,10,11,12]. Since the hydrophobic pore of the carbon nanotubes is slightly similar to the biological ion channels with intrinsic ion selectivity [13,14], the CNTs have the potential to act as reverse osmosis membranes for water treatment. However, it still faces problems such as the tendency of agglomeration and insolubility in various solvents and different polymers, together with the weak interaction between the carbon nanotubes and the polymer matrix.

To enhance the solubility and compatibility of the CNTs in various organic solvents and polymers, different strategies have been reported, such as the oxidation of carbon nanotubes by HNO_3_ [15,16], H_2_SO_4_/HNO_3_ [17,18,19,20,21,22,23,24,25,26,27,28,29,30], O_3_ [31], K_2_Cr_2_O_7_ [32], H_2_O_2_ [33], etc.; fluorination [34,35] and plasma treatments [36,37]; radiation-induced graft polymerization process [38]; and further substitution reaction [17,18,39,40,41,42,43], for example, grafting p-aminobenzenesulfonic acid, straight-chain alkyl-diamines, 3-aminopropyltriethoxysilane, and so on. In addition, the effects of the grafting rate on CNTs’ dispersion and stability [15,20,22] have also been explored.

Moreover, to obtain better performance of the polyamide membranes, various methods have been adopted to further improve the water flux, desalination, and antifouling properties of the polyamide membrane. For example, surface-modified, zeolite-polyamide membranes were prepared by Hang et al. [44]. The new PES-based, mixed-matrix membranes using polycaprolactone as an additive to modify the CNTs were fabricated by Mansourpanah et al. [45]. Zhang et al. prepared mixed-acid-treated, multi-walled, carbon nanotube-polyamide membranes [3]. In addition, many researchers have modified the CNTs and added it to the polyamide membranes by interfacial polymerization to improve the water flux, desalination rate, and antifouling properties [26,27,28,29,46,47,48].

In view of this, our research group conducted a series of simulations of modified carbon nanotubes for desalination [49,50]. Results showed that 100% salt rejection could be achieved with high water flux when certain types and numbers of functional groups (-COOH, -CONH_2,_ and -NH_2_) were added to the tip and/or interior of single-walled CNTs with diameters of 1.356 nm or 1.763 nm. For CNTs with a diameter greater than 2 nm, long-chain groups need to be added to achieve 100% salt rejection. Thus, our research group further implemented experiments by grafting modification (grafting -COOH, -COCl, -CONH_2,_ and -NH_2_) groups to 0.9 nm and 3.5 nm SWCNTs and 4.9 nm (average diameter) MWCNTs. The experimental results showed that the dispersibility and stabilization of the functionalized CNTs were significantly improved, and the water flux and desalination rate of the SWCNT- and MWCNT-doped PA membranes were better than the PA membrane. However, the salt rejection rate of the short-chain modified MWCNTs’ membranes was lower than that of the SWCNTs’ membranes, which may be related to the MWCNTs having a larger diameter. Therefore, we speculate that grafting long aliphatic chains on MWCNTs might further improve water flux and salt rejection. In this paper, long aliphatic chains (12-chloro-12-oxododecanedioic acid-methyl ester, containing two ester groups, could make water molecules form hydrogen bonds, which is expected to improve the hydrophilicity of MWCNTs) were grafted on MWCNTs to explore the effects of water flux and the desalination rate of membranes.

## 2. Experimental Section

### 2.1. Chemicals and Solvents

Multi-walled carbon nanotubes (MWCNTs, the diameter distribution of MWCNTs was shown in Figure 1c, the diameter of the largest distribution from 3 nm to 7 nm, length 10–30 μm) were manufactured by Beijing gold deco island technology Co., Ltd., Beijing, China. Sulfuric acid, hydrochloric acid, nitric acid, dimethylformamide (DMF), lithium aluminum hydride (LiAlH_4_), pyridine (Py), tetrahydrofuran ethyl acetate, triethylamine, dimethyl sulfoxide, m-phenylenediamine (MPD, 99%), dodecanedioic acid (99%), trimesoyl chloride (TMC, 98%), (±)-camphor-10-sulfonic acid, sodium dodecyl sulfate, anhydrous toluene, oxalyl chloride, sodium hydrogen carbonate (NaHCO_3_), sodium chloride, anhydrous magnesium sulfate, potassium hydroxide, methanol, ethanol, n-hexane (97%), polysulfone (PSF) membranes, and polytetrafluoroethylene (PTFE) membranes (pore size 0.22 μm, diameter 50 mm) were used as received (Appendix A).

### 2.2. Oxidation of Multi-Walled Carbon Nanotubes

Pristine MWCNTs (P-MWCNTs) were soaked in 60 mL of mixed acid (H_2_SO_4_/HNO_3_ = 3/1 by volume) and then stirred for 4 h at reflux (60 °C). The obtained MWCNTs solution was diluted by de-ionized water until the pH of the filtrate became neutral and then filtered by the PTFE membrane to obtain acidified MWCNTs (MWCNT-COOH). Then, the obtained MWCNT-COOH was dried for 12 h at 60 °C in a vacuum-drying oven.

### 2.3. Reduction of Acidified Carbon Nanotubes

Tetrahydrofuran (50 mL), MWCNT-COOH (160 mg), and LiAlH_4_ (100 mg, 0.0026 mol, as reducing agents) were added into a flask. After reacting for 5 h under ice-bath conditions and with N_2_ as the protective gas, the products were filtered with a PTFE membrane and washed with ethanol, diluted hydrochloric acid, and water, in turn. Finally, the reduction products were dried for 12 h (60 °C) in a vacuum-drying oven.

### 2.4. Synthesis of Dimethyldodecanedioate

Dodecanedioic acid (20 g, 0.01 mol, 99% purity) was dissolved in 100 mL of methanol solution, to which 2 mL of concentrated sulfuric acid was added. After stirring for 24 h at a reflux (70 °C), the solvent was distilled in a rotary evaporator, and the distillation residue was directly poured into ice water, then subsequently extracted with three portions of ethyl acetate. The combined organic layers were first washed with 10% NaHCO_3_ solution and then washed with a saturated salt solution. Finally, the filtrate was poured into the rotatory evaporator to remove the excess organic solvent. The dimethyldodecanedioate was obtained as a colorless oily liquid (product: 21.074 g, 0.0817 mol, yield: 93.93%).

### 2.5. Synthesis of Dodecanedioicacid-1-Methylester

Potassium hydroxide (5.38 g, 0.096 mol, 85%) and dimethyldodecanedioate (21.074 g, 0.0817 mol) were dissolved in 200 mL of methanol solution and kept at room temperature with 4 h of continuous stirring. Then, the solvent was distilled in a rotary evaporator to remove the solvent methanol and obtain the white product. The distillation residue was dissolved in ethyl acetate, and the organic and aqueous layers were separated by extraction. The aqueous layer of the product was acidified by hydrochloric acid (36%) to pH = 3, then extracted with ethyl acetate three times. The organic-layer product was washed by saturated salt water, evaporated to concentrate and recycle the dimethyldodecanedioate, and then dried with anhydrous magnesium sulfate and filtered. The solvent (ethyl acetate) was removed by evaporation to obtain a white oil mixture, which was completely cooled by treating it with hexane, and then subsequently stratified. The lower layer was a white, oily liquid, and the upper layer was a colorless liquid. The hexane was filtered and evaporated to obtain the dodecanedioicacid-1-methylester with the colorless, oily liquid (product: 4.9045 g, 0.0201 mol, Yield: 93.39%).

### 2.6. Synthesis of 12-Chloro-12-Oxododecanedioic Acid-Methyl Ester

The intermediate product dodecanedioicacid-1-methylester (4.9045 g, 0.0201 mol) was dissolved in 40 mL of anhydrous toluene under 5 °C ice baths. Then, two drops of DMF and 5 mL of oxalyl chloride was added in turn. Reduced-pressure distillation was used to obtain a pale-yellow, oily liquid. The resulting solution of 12-chloro-12-oxododecanedioic acid-methyl ester was directly used in the next step. The synthesis scheme of the product 12-chloro-12-oxododecanedioic acid-methyl ester is shown in Figure 2.

### 2.7. Grafting Reaction of MWCNTs with 12-Chloro-12-Oxododecanedioic Acid-Methyl Ester

The product of 12-chloro-12-oxododecanedioic acid-methyl ester (5.4331 g, 0.0207 mol), mixed solvent (15 mL DMF and 15 mL pyridine), and 200 mg of MWCNT-CH_2_OH were added into the round-bottomed flask. After reacting for 2 h in ice water, the product MWCNT-CH_2_OCOC_12_H2_3_O_3_ was filtered, washed with water and ethanol, and dried for 12 h at 60 °C.

### 2.8. Fabrication of Polyamide Membrane and MWCNT-Polyamide Membrane

A total of 0.05 g MWCNTs (P-MWCNTs, MWCNT-CH_2_OCOC_12_H_23_O_3_) were dispersed in 100 mL of 2.0 wt% MPD solution to prepare the 0.05 wt% concentration, CNT-dispersed solution. The PSF ultrafiltration membrane was attached to a mold and washed with ultrapure water several times to remove the impurities, followed by blowing off the moisture and bubbles on the surface of the membranes with N_2_. Then, 100 mL of 2.0 wt% MPD solution, 0 wt% MWCNTs, and 0.05 wt% of different types of MWCNTs (P-MWCNTs, MWCNT-CH_2_OCOC_12_H_23_O_3_) were, respectively, cast on the top surface of the PSF substrate and kept in the mold for 5 min to ensure that the MPD solution penetrated into the pores of the substrate. Finally, the MPD solution was poured out, and then the residual solution on the membrane’s surface was removed by N_2_. The n-hexane containing 0.2 wt% TMC solution was poured onto the surface of the substrate for 30 s for interfacial polymerization. Then, the oil phase was poured out, and the produced membrane was placed in the oven at 80 °C for 10 min. The polyamide membranes containing different types of MWCNTs are called PA, PA-MWCNTs, and PA-MWCNT-CH_2_OCOC_12_H_23_O_3_, respectively.

### 2.9. Characterization of MWCNTs and Polyamide Membrane

The P-MWCNTs and functionalized MWCNTs were characterized with TEM (JEM 2100F, JEOL, Tokyo, Japan) and FT-IR (Nicolet IR200, AFN0800865, Thermo Scientific, Waltham, MA, USA). The graft ratio was characterized using a TGA (HCT-1, 12-036). The performance of the membranes was characterized by SEM (JEOL 7500F, JEOL Oxford (EDS) Co., Tokyo, Japan), contact angle goniometer (DSA100, KRüSS, Schleswig-Holstein, Germany), and a membrane-performance evaluation instrument (GY70-6, Hangzhou Water Treatment Technology Research and Development Center Co., Ltd., Hangzhou, China).

### 2.10. Membrane Filtration Test

The filtration experiment of the membrane was carried out through the membrane-performance evaluation instrument, with the effective filtration area of 20.418 cm^2^. The performance of the RO membrane was investigated by a cross-flow flat sheet. The schematic filtration of the RO membrane for desalination is shown in Figure 1. The feed liquid was 2000 mg/L of NaCl solution (pH = 8), and the conductivity was 3.0 mS/cm; the pressure was maintained at approximately 1.0 MPa, and the temperature was controlled at 25 °C. In order to optimize the test conditions of the membrane, the flux of the membrane was measured after a stabilization stage for 90 min at the operating pressure. Then, after 10 min, the permeated solution was collected, weighed, and measured for the electrical conductance. The membrane flux (*J*) was calculated according to Equation (1):(1)$$J=\frac{\Delta V}{A\times \Delta t}$$
where Δ*V* is the volume of the permeate collected by the membrane in 10 min, *A* is the effective surface area of the membrane, and Δ*t* is the time needed to collect the permeate. Salt rejection (*R*) was calculated according to Equation (2):(2)$$R=(1-\frac{C_{p}}{C_{f}})\times 100\%$$
where *R* is the parameter of the salt rejection, *C_p_* is the salt concentration in the permeate, and *C_f_* is the salt concentration in the feed.

## 3. Results and Discussion

Figure 3a,b represents the Fourier-transform infrared (FT-IR) analysis of dodecanedioic acid and 12-chloro-12-oxododecanedioic acid-methyl ester, respectively. The main absorption peaks of the groups are marked in Figure 3. Among them, in Figure 3a, the peak at 928 cm^−1^ represents the out-of-plane bending of -OH, the peaks at 1294 cm^−1^ and 1411 cm^−1^ correspond to the stretching-vibration peak of C-O and the in-plane-bending peak of C-H, respectively, and the peaks at 2853 cm^−1^ and 2921 cm^−1^ represent the symmetric and asymmetric stretching vibrations of the methylene group (-CH2-), which indicates the presence of dodecanedioic acid. For Figure 3b, a new peak at 725 cm^−1^ indicates the presence of C-Cl. The peak at 1172 cm^−1^~1200 cm^−1^ is C-C(=O)-C, the peak at 1740 cm^−1^~1803 cm^−1^ is the stretching vibration of C=O, and the peak at 2857 cm^−1^ is the methyl peak, which proves the presence of COCl and –COOMe. By comparing Figure 3a,b, it can be seen that the methyl 12-chloro-12-oxydodecanedioate was successfully synthesized.

Figure 1a,b represents the transmission-electron-microscope images of the P-MWCNTs and MWCNT-CH_2_OCOC_12_H_23_O_3_, respectively. Figure 1c,d shows a Gaussian distribution curve of the inner diameter of MWCNTs, measured by ImageJ software, corresponding to Figure 1a,b. As shown in Figure 1a, it is clear that the pristine MWCNTs were unevenly dispersed in the aqueous solution, and almost all of the original MWCNTs were closed at the ends. After being modified (Figure 1b), the MWCNTs could be well-dispersed in the aqueous solution, with loose morphology and most ports of the carbon tube opened (Appendix A). In addition, there was no obvious change in the external morphology of the carbon tubes after chemical treatment, which indicates that the current conditions are not disruptive to the external morphology of the tubes. As shown in Figure 1c,d, the inner diameter (larger distribution of diameter) of the MWNTs was obviously decreased after the functionalizing process. The results indicate that the 12-chloro-12-oxododecanedioic acid-methyl ester has been successfully grafted into interior of the CNTs. The simulation results show that the minimum value of the 12-chloro-12-oxododecanedioic acid-methyl ester is 2.063 nm (as shown in Figure 4, 0.5 ns minimize calculation, 1 ns NVT ensemble calculation under 300 K by NAMD software), consistent with the inner diameter of MWCNTs modified by 12-chloro-12-oxododecanedioic acid-methyl ester, which further indicates that the 12-chloro-12-oxododecanedioic acid-methyl ester has been successfully grafted on the inner wall of the MWCNTs.

Figure 5 shows the TGA thermograms of MWCNTs (a program of 10 °C/min to 800 °C in a mixture of nitrogen and oxygen as the protective gas). Figure 5a–d represents the pristine MWCNTs, MWCNT-COOH, MWCNT-CH_2_OH, and MWCNT-CH_2_OCOC_12_H_23_O_3_, respectively. The weight-loss rates of the four curves were different, indicating that the functional groups grafted on multi-walled carbon nanotubes affect the stability of MWNTS. Compared to the original MWCNTs, the thermal stability of the grafted MWCNTs was given a varying influence, especially with the MWCNT-CH_2_OH. The reason is that the -COOH binds to MWCNTs more stably than the -OH. The MWCNT-CH_2_OCOC_12_H_23_O_3_ also shows better thermal stability than the MWCNTs-CH_2_OH.

Figure 6a–c shows the SEM images of the membrane surface morphology of the membranes of PA, PA-MWCNTs, and MWCNTs-CH_2_OC_12_H_23_O_3_, respectively. From the morphologies of the above three composite films, it can be seen that the surface morphologies are basically similar. In addition, the MWCNTs-CH_2_OC_12_H_23_O_3_-polyamide membrane presented more nodes than the PA membrane, due to the relevant stability and dispersion of the multi-walled carbon nanotubes on the surface of the membrane. This difference in the surface morphology of MWCNT-polyamide composites may be related to the fact that the MWCNTs modified with methyl 12-chloro-12-oxydodecanedicarboxylate groups have hydrophilic functional groups such as ester groups, and they have good conductivity.

Based on the above results, we calculated the hydrophilic nature of the three membranes by the water contact angle (Figure 7). As shown in Figure 7, the contact angle of the MWCNT-polyamide membranes was much lower than that of the PA membrane, especially since the hydrophilicity of the PA-MWCNT’s membrane was significantly improved after the addition of the long-chain functional group 12-chloro-12-oxododecanedioic acid-methyl ester at the inner wall of the MWCNT. The decrease in the contact angle was ascribed to the reduction of the surface roughness of membranes and the embedded hydrophilic groups, such as the carboxyl groups and 12-chloro-12-oxododecanedioic acid-methyl ester. With the change of additive content, the obvious change of contact angle is not only related to the variation of membrane-surface roughness but also the hydrophilicity of the functional group. The looseness of the membranes’ surface structure is the main factor for the decrease in WCA value.

The water flux and salt rejection of a membrane are the two most important features in determining water desalination performance, and thus, it is necessary for them to be characterized. Figure 8 shows the water flux and salt rejection of the PA membrane and PA-CNT-composite membranes (the concentration of NaCl feed solution was 2000 ppm, the feed pressure was 1.0 MPa). The values in Figure 6 are the averages of the three membrane samples prepared at different times (at least three samples were measured for each membrane). As shown in Figure 8, compared to the bare PA membrane, the water flux and salt rejection rate of the MWCNT-polyamide membrane increased significantly, especially the MWCNT modified with 12-chloro-12-oxododecanedioic acid-methyl ester groups. (Compared with the bare PA membrane, the water flux of the membrane with MWCNT-CH_2_OCOC_12_H_23_O_3_ increased 21.7%, and the rate of salt rejection increased 5.07%). The long-chain functional groups of 12-oxododecanedioic acid-methyl ester contain two ester groups, and the -COOCH3 terminal group can make water molecules form hydrogen bonds more easily and allow the admission and passage of water, which is expected to improve the hydrophilicity of MWNTs. To prove our hypothesis, we performed 1-ns molecular dynamics simulations, and the statistics of the hydrogen bond distribution (Appendix A) of MWCNT and MWCNT-CH_2_OCOC_12_H_23_O_3_ were plotted for analysis. The results showed that MWCNT-CH2OCOC_12_H_23_O_3_ exhibited more hydrogen bonding (Appendix A). The contact angle measured in Figure 7 also proves the enhancement of the hydrophilicity of the MWCNT-CH_2_OCOC_12_H_23_O_3_ membrane. At the same time, the structure of smaller interior diameters observed for the modified MWCNT, similar to aquaporins in living organisms, accelerated the passage of water. (Since the molecular diameter of a water molecule is about 2.75 Å and the diameter of MWCNTs-CH_2_OCOC_12_H_23_O_3_ are close to the 28 Å, there is plenty of room for these to pass through the space between the long-chain esters and the inner walls of the CNT, even within the reduced space in the bottle-necked formations). On the other hand, the functionalization also occurred on the outer walls, which could increase the repulsions between the terminal -COOCH_3_ group and vicinal CNTs, which made the water molecules more accessible. Therefore, the water flux was increased. Similarly, the attraction electrostatic interaction of sodium ions and the repulsion electrostatic interaction of chloride ions by ester groups of 12-chloro-12-oxododecanedioic acid-methyl ester also increased the desalination rate. The comparison between the materials on the two major performance indicators demonstrates that MWNTs modified with 12-chloro-12-oxododecanedioic acid-methyl ester groups can desalinate water at a higher rate than PA membranes.

## 4. Conclusions

In this work, MWCNTs were modified with the long aliphatic chain 12-chloro-12-oxododecanedioic acid-methyl ester on the entrance and inner walls, and then the functionalized CNTs were added to the polyamide layer to prepare the MWCNT-CH_2_OCOC_12_H_23_O_3_ membranes for desalination. The characterization by FT-IR, TEM, and TGA confirmed that the 12-chloro-12-oxododecanedioic acid-methyl ester groups were successfully grafted into the MWCNTs. The SEM images showed that the morphology of the modified membranes was more homogeneous due to the hydrophilicity of the membranes that were improved after grafting the 12-chloro-12-oxododecanedioic acid-methyl ester. The membrane-performance-evaluation experiments showed that the water flux and desalination rates of the MWCNT-polyamide membrane were both improved compared to the bare PA membrane, especially the MWCNTs modified with 12-chloro-12-oxododecanedioic acid-methyl ester groups, due to the fact that 12-chloro-12-oxododecanedioic acid-methyl ester groups could form hydrogen bonds with water and had electrostatic interaction with sodium and chloride.

## Data Availability

The data presented in this study are available on request from the corresponding author.

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
