# Peer review of "Long-Chain Modification of the Tips and Inner Walls of MWCNTs and Their Nanocomposite Reverse Osmosis Membranes"

_membranes, 2022, doi:10.3390/membranes12080794_

Round 1
Reviewer 1 Report
Dear authors,
Please see carefully the following points:
Comments on Μembranes-1829599
Title: “Long chain modification of 4.9 nm MWCNTs on tip and inner 2 wall and its nanocomposite reverse osmosis membrane”
By Qing Li, Dengfeng Yang, Jianhua Wang, Qingzhi Liu, Zhun Ma and Dongmei Xu.
College of Chemical and Biological Engineering, Shandong University of Science and Technology,
College of Chemistry and Pharmaceutical Science, Qingdao Agriculture University, Qingdao City 266109, 6 Shandong, China
Reviewer’s Comments:
1. The tile must be shorter, more condense and the described size of 4.9nm must be removed
2. I suggest to the authors to replace this “4.9nm diameter”. The used CNTs present a fluctuation on their diameter sizes, as it is obvious from Fig. 3 and it is mistake to provide the value of the wide peak distribution as the characteristic size of the used materials.
3. Fig. 7 must be presented in a clearly way concerning which picture is what. Please separate the three images, A, B and C.
4. The authors must explain why they chosen to modify the MWCNTs by the presented two ways.
5. Information concerning the differences in the CNTs’ dispersion among the three solutions, these of the PA, PA-MWCNTs and PA-MWCNT-H14H25O4
6. Fig.5 presents the TGA curves for the samples (a) MWCNTs, (b) MWCNT-COOH,. (c) MWCNT-OH, (d) MWCNT-C14H25O4. However, only the two of them are used as PA membrane’s fillers. Why the other didn’t use?
7. The authors reported the water flux and salt rejection of PA membrane and PA-CNT composite membranes. (2000ppm of NaCl feed solution, 1.0 MPa of feed pressure). As it is mentioned in comment 6 we don’t have any results for the other 2 membranes that we expected to be prepared and tested. In addition, more information about the reverse osmosis (RO) experimental conditions must be included in the manuscript.
8. More analysis is needed for the Fig.5. For example, why the weight loss of the sample (b) is recorded smaller than the (c). Which is the explanation? Any conclusion about the grade of the CNTs functionality etc?
9. The results referred to Fig. 4 must be presented clearly and relevant literature must be added and discussed.
1- Fig. 2 also presents the FTIR spectra of (a) dodecanedioic acid, (b) 12-chloro-12-oxododecanedioic acid. I can’t understand the reason of these measurements. Actually, in the manuscript JUST described which are the characteristic peaks for each spectrum ((a) and (b)) BUT the authors didn’t discuss anything other…
1- The water flux and salt rejection of PA membrane and PA-CNT composite membranes must be compared with similar materials from the literature. I think that the achieved values are not in a highly record level compared with the literature… If this is correct the authors must be explain finally what the novelty of their work is?
1- Any explanation about the differences of the WCA measurement?
--- Overall your work needs remarkable changes in order to achieve the standards of membranes.
Author Response
Response to Reviewer 1
For Reviewer 1:
Comments:
Reviewer #1:
1.The tile must be shorter, more condense and the described size of 4.9nm must be removed.
Authors’ reply: Thank you for your valuable opinion. The tile had been shortened and replaced by “Long-chain modification of the tips and inner walls of MWNTs and their nanocomposite ROs membranes”.
- I suggest to the authors to replace this “4.9nm diameter”. The used CNTs present a fluctuation on their diameter sizes, as it is obvious from Fig. 3 and it is mistake to provide the value of the wide peak distribution as the characteristic size of the used materials.
Authors’ reply: We thank the reviewers for this constructive comment. We have deleted “4.9nm diameter” in tile, abstract and conclusion, and added explanation in line 82~83 of Method part (the diameter distribution of MWCNTs was shown in Fig.3c, the diameter of the largest distribution was 4.9 nm), line 69 of Introduction part (average diameter) and 210 line of Results and discussion part (larger distribution of diameter) with blue color.
- Fig. 7 must be presented in a clearly way concerning which picture is what. Please separate the three images, A, B and C.
Authors’ reply: Thank you for pointing this out. We have described in detail in the fig.7 caption and separate the three images, A, B and C.
Figure 7. The SEM images of surface morphology of (A) PA membranes, (B) PA and MWCNTs nanocomposite membrane, (C) PA and 12-oxododecanedioic acid(C14H25O4) modified MWCNTs nanocomposite membrane.
- The authors must explain why they chosen to modify the MWCNTs by the presented two ways.
Authors’ reply: We thank the reviewers for this constructive comment. Our research group used molecular dynamics simulation methods to find that adding these two modified groups to the inner wall and port of carbon nanotubes could improve the desalination rate (Desalination and Water treatment,101(2016)61-69), so this paper considers adding long-chain modified groups to the ports and inner walls of carbon nanotubes.
- Information concerning the differences in the CNTs’ dispersion among the three solutions, these of the PA, PA-MWCNTs and PA-MWCNT-C14H25O4.
Authors’ reply: We thank the reviewers for their comments. In order to determine the effect of chemical modification on the dispersibility and stability of carbon nanotubes, the water dispersibility of MWNTs was studied by static sedimentation method. MWCNT, MWCNT-COOH, MWCNT-OH, MWCNT-COCl, MWCNT-C14H25O4 was dispersed in water, sonicated for 30 min, and allowed to stand for a while. Fig.S1 lists the water dispersion of the samples after 0 h (a), 1 h (b), and 40 weeks (c) of ultrasonic standing (samples from left to right are MWCNT, MWCNT-COOH, MWCNT-CH2OH, MWCNT-COCl, MWCNT-C14H25O4). As shown in Fig.S1, the MWCNTs began to coagulate and precipitate after 1 hour of ultrasonic standing, while the MWNTs with grafted functional groups still maintained good dispersibility and stability after ultrasonic storage for 40 weeks or even longer.
FigS1 Dispersion and stability of samples after ultrasonic 0 h (a), 1 h (b), 40 weeks (c), (samples from left to right are MWCNT, MWCNT-COOH, MWCNT-CH2OH, MWCNT-COCl , MWCNT- CH2OCOC12H23O3).
- Fig.5 presents the TGA curves for the samples (a) MWCNTs, (b) MWCNT-COOH, (c) MWCNT-OH, (d) MWCNT-C14H25O4. However, only the two of them are used as PA membrane’s fillers. Why the other didn’t use?
Authors’ reply: We thank the reviewers for their construction comments. In this paper, our main research objective was the effect of MWCNTs modified by long-chain modified groups on the permeability of MWCNTs/PA nanocomposite membranes. In the experiment, MWCNT-COOH and MWCNT-CH2OH were applied as the mid product in the preparation of long chains MWCNT-C14H25O4. The TGA data of MWCNT-COOH and MWCNT-OH were used to compare the difference between materials and product. So only two cases of MWCNTs and MWCNT-C14H25O4 membranes were considered.
- The authors reported the water flux and salt rejection of PA membrane and PA-CNT composite membranes. (2000ppm of NaCl feed solution, 1.0 MPa of feed pressure). As it is mentioned in comment 6, we don’t have any results for the other 2 membranes that we expected to be prepared and tested. In addition, more information about the reverse osmosis (RO) experimental conditions must be included in the manuscript.
Authors’ reply: We thank the reviewers for their construction comments. The MWCNT-COOH and MWCNT-OH were the mid product in the preparation of long chains MWCNT-C14H25O4, thus the two membranes were not prepared and measured. And the schematic filtration of RO membrane for desalination had been added in the manuscript.
- More analysis is needed for the Fig.5. For example, why the weight loss of the sample (b) is recorded smaller than the (c). Which is the explanation? Any conclusion about the grade of the CNTs functionality etc?
Authors’ reply: We thank the reviewers for their construction comments. The weight loss rates of the four curves were different, indicating that functional groups were grafted on the multi-walled carbon nanotubes and changing the stability performance. Compared with original MWCNTs, the thermal stability of grafted MWCNTs were given a varying influence, especially the MWCNT-OH. The reason is that the -COOH binds to MWCNTs more stable than the -OH. And the MWCNT-C14H25O4 also shown better thermal stability than MWCNT-OH.
- The results referred to Fig. 4 must be presented clearly and relevant literature must be added and discussed.
Authors’ reply: We thank the reviewers for their construction comments. We add some calculation detail and discussed of Fig.4.
The simulation results shown that the minimum value of the 12-oxododecanedioic acid, was 2.063 nm (as shown in Fig.4, 0.5ns minimize calculation 1ns NVT ensemble calculation under 300K by NAMD software), consisting with inner diameter of MWCNTs modified by 12-chloro-12-oxododecanedioic acid-methyl ester, which further indicated that the 12-oxododecanedioic acid has been successfully grafted on the inner wall of MWCNTs.
1- Fig. 2 also presents the FTIR spectra of (a) dodecanedioic acid, (b) 12-chloro-12-oxododecanedioic acid. I can’t understand the reason of these measurements. Actually, in the manuscript JUST described which are the characteristic peaks for each spectrum ((a) and (b)) BUT the authors didn’t discuss anything other…
Authors’ reply: We thank the reviewers for their construction comments. In this paper, dodecanedioic acid was the materials of preparation 12-chloro-12-oxododecanedioic acid-methyl ester, thus, the FTIR spectra of dodecanedioic acid and 12-chloro-12-oxododecanedioic acid were presented. These measurements were supported to demonstrate the successful synthesis of the 12-chloro-12-oxododecanedioic acid-methyl ester. The peaks at 928 cm-1 represented the out of plane bending of -OH. The peaks at 1294 cm-1 and 1411 cm-1 corresponded to the C-O stretching vibration peak and the inner plane bending of C-H, respectively. The peaks at 2853 cm-1 and 2921 cm-1 represent the symmetric and asymmetric stretching vibrations of methylene (-CH2-). In the Fig.2b, a new peak C-Cl appeared at about 725 cm-1. The peak in the range of 1172 cm-1 to 1200 cm-1 represented the C-COO-, the peak at the peak value of 1740 cm-1 to 1803 cm-1 represented the stretching vibration of C=O. Compared with Fig.2a and Fig.2b, it was shown that 12-chloro-12-oxododecanedioic acid has been successfully synthesized.
1- The water flux and salt rejection of PA membrane and PA-CNT composite membranes must be compared with similar materials from the literature. I think that the achieved values are not in a highly record level compared with the literature… If this is correct the authors must be explain finally what the novelty of their work is?
Authors’ reply: We thank the reviewers for their construction comments. In this paper, the modification of MWCNTs was in the interior and at the entrance simultaneously, instead of traditional port modification. Our previous molecular simulation researches showed that adding modified groups to the inner wall and port of carbon nanotubes could improve the desalination rate (Desalination and Water treatment,101(2016)61-69) than only port-modified. ( Applied Surface Science,496 (2019) 143680), the water flux and salt rejection of membrane with MWCNT-C14H25O4 shown an obvious reduction. The reason may due to many factors such as CNT content, materials of basic membrane, experiment conditions, etc. Under the same conditions, the water flux and salt rejection of membrane with MWCNT-C14H25O4 shown an obvious improvement.
1- Any explanation about the differences of the WCA measurement?
Authors’ reply: We thank the reviewers for their construction comments. Fig.8 showed the hydrophilic nature of MWCNT-polyamide membrane by the water contact angle. It can be seen from the graph that the contact angle of the MWCNT-polyamide membranes was much lower than that of the PA membrane, especially the addition of the long chain functional groups12-oxododecanedioic acid at the inner wall of the MWCNTs, which improved the hydrophilicity of the PA-MWCNTs membrane. The decrease of the contact angle was ascribed to both the reduction of the surface roughness of membranes and the embedded hydrophilic groups such as carboxyl groups and 12-oxododecanedioic acid. With the change of additive content, the obvious change of contact angle is not only due to the variation of membrane surface roughness, but also the hydrophilicity of the functional group. As the main influence, loose structure of membrane surface leads the decrease of WCA.

Reviewer 2 Report
Dear Sir,
The manuscript is reporting the use of a functionalized MWCNT as useful component in a membrane for desalination uses. The authors introduced a long hydrocarbonated chain in order to facilitate the passage of purified water through the membrane. The particularity and originality of the manuscript resides in the particular additive used: 12-chloro-12-oxododecanedioic acid-methyl ester. One aspect that would need some clarification in the Introduction paragraph is why exactly this compound? Why not using a simple fatty acid acyl chloride in the reaction with MWCNT-CH2-OH to obtain a functionalized MWCNT in which the hydrocarbonated chain is linked through an ester function or a fatty amine in the reaction with MWCNT-COOH (in which the hydrocarbonated chain is linked through an amide function)? What would be the expected role of the terminal -COOMe group?
What is the irrefutable indication the the functionalization occurred inside the CNT and not on the outer walls, which would be more probable?
Why the authors assume throughout the text the presence of a 12-oxododecanedioic acid, when the synthesis was stoped at an ester moiety? There are no indication of further hydrolysis. Moreover, the formula used by the authors corresponds to an ester.
The IR spectra need a better description. Fig 2b is showing the formation of the monoester acylchloride more clearly through the peaks at 1740 cm-1 and 1803 cm-1 which represent the stretching vibration of carbonyl C=O bond specific to acyl chlorides (1803), respectively ester (1740). The C=O absorption of -COOH group is more toward the 1700s.
The presence of a single 12-oxododecanedioic acid-methyl ester strain inside the CNT is, in my opinion, not enough to “clog” the CNT and explain the reduced “diameter” (in fact, according to authors’explanation, in would be in fact a “half-diameter”). I suggest another possible explanation for the reduced diameter: imagine that not one, but two such long chained substitutents face each-other on opposite sides of the inner wall; could atractions such hydrogen bonding (there are -COOCH3 moieties at each end) or hydrophobe-hydrophobe interactions between the hydrocarbonated long chains could explain the reduction in diameter?
A problem is that the authors assume that: “Due to addition of the long chain functional groups of 12-oxododecanedioic acid at the inner wall of the CNTs, the hydrophobic CNTs were converted into hydrophilic ones, which made the water molecules more accessible.” I disagree with that statement: first the moiety introduced is not an acid, but an ester. Second, it has a long hydrocarbonated chain, which should increase the hydrophobicity (not the hydrophilicity). Third, the inner diameter of the MWCNTs is smaller, according to the authors’ own measurments. I suggest another possible explanation: the -COOCH3 terminal group allows the admission and passage of water. In the same time, the smaller diameters observed for the modified MWCNT does not represent a shrinked CNT, but shows that inside these there some kind of bottle-neck formations – these could act as some kind of miniature Venturi tubes that accelerates tha passage of water. Since the molecular diameter of a water molecule is about 2.75A (27.5 nm – interestingly close to the 28 nm inner diameter of the modified MWCNTs) there is plenty of room for these to pass through the space between the long-chain esters and the inner walls of the CNT, even within the reduce space in the bottle-necked formations. On the other hand, if the functionalization occurred on the outer walls, could the pressure of a large group, associated with repulsions between the terminal -COOCH3 group and vicinal other CNTs explain the diminished inner diameter?
General comments:
- The text should be re-edited for proper English language and clearer presentation. I wil give only one example, but the entire manuscript must be corrected.
“20g dodecanedioic acid (99%) was dissolved in 100 mL methanol solution and then 2 mL concentrated sulfuric acid was added. After being treated for 24 h by stirring reflux at 70℃, the solvent was distilled by a rotary evaporator, and the distilled residues were poured into the ice water, then extracted with ethyl acetate for three times and combined with organic layer which was washed with 10% sodium hydrogen carbonate and saturated salt water in turn. In the end, after drying with anhydrous magnesium sulfate and filtering, the filtrate was poured into the rotator evaporator to remove the solvent ethyl acetate. Subsequently, the colorless oily liquid dimethyldodecanedioate was obtained.” - “Dodecanedioic acid (20 g, 99% purity) was dissolved in 100 mL methanol solution after which 2 mL concentrated sulfuric acid were added. After stirring for 24 h at reflux (70℃), the solvent was distilled in a rotary evaporator, and the distillation residue was directly poured into ice water, subsequently extracted with three portions of ethyl acetate. The combined organic layers were first washed with 10% NaHCO3 solution and then with a saturated salt solution. Finally, after drying on anhydrous magnesium sulfate and subsequent filtration, the filtrate was poured into the rotatory evaporator to remove the excess organic solvent. Thus, the brute dimethyldodecanedioate was obtained as a colorless oily liquid.”
- In the Experimental part, in the description of the syntheses, give also the moles in brackets for each reagent – it will help to establis the molar ratio between the reagents for each step of the process, since the synthesis implies a complete esterification, followed by a statistical mono-hydrolysis (be the way, how the authors can be sure that by-products such as dodecanedioic acid and unreacted dimethyldodecanedioate were also present in the reaction mixture?)
- there is a discrepency between Figure 1 (the synthesis steps) and the nomenclature used: what exactly is the end point of the synthesis, the acid (as in the text) or the ester (as in Fig.1). In other words, is the moiety in C1 a -COOH or a -COOCH3 group? From the general formula used for the functionalized MWCNT I assume it is the ester, but the authors should clarify this and therefore use the proper names in the text.
- I assume that the long chain oxododecanedioic acid is grafted on the MWCNTs through a SN2 reaction between the -CH2OH moiett and the acyl chloride. Therefore, I think it would be more suitable, since the starting material was designated as MWCNT-CH2-OH, to write the reaction product as MWCNT-CH2-O-CO-C12H23O2. For example, it would be clearer in a lict such as “MWCNTs, MWCNT-COOH, MWCNT-OH and MWCNT- 220 C14H25O4”, which would become “MWCNTs, MWCNT-COOH, MWCNT-CH2-OH and MWCNT-CH2-OCO-C12H23O3”.
- “The product of 12-chloro-12-oxododecanedioic acid was added into the round bottom flask, then 15 mL DMF and 15 mL pyridine were mixed as solvent and acid binding agent, and then 200 mg MWCNT-CH2OH were added. After keeping a continuous reaction for 2 h in ice water, the product MWCNT-C14H25O4 was filtered and then washed with water and ethanol separately and dried at 60℃for 12 h.” – what was the amount of ester used in the functionalization process? What was the amount of fatty ester grafts introduced on the MWCNTs? Eventually this could be determined by knowing the starting quantity of 12-chloro-12-oxododecanedioic acid methyl ester used and by measuring the quantity found in the solution after filtration.
Minor comments:
- separation of “polymer” is not “pol – ymer”, but “po-lymer” or “poly-mer”; same observation for “properties” – “pro-perties” or “proper-ties”; same observation many other strange such separations
- use “mL” as IS symbol for milliliter
- “water layer” – “aqueous layer”
- “stratrificated” ???
- the correct name of the end-product of the synthesis is not “12-chloro-12-oxododecanedioic acid”, but “12-chloro-12-oxododecanedioic acid-methyl ester”
- leave some blank space between figures 3a and 3b; same for fig. 7a-c
- fig 4 presents a length of 20.63 nm for the hydrocarbonated -CH2-O-CO-C10H20-COOCH3 chain, while in the text the value that is given is 2.063 nm; which is correct? I assume one is in nm and the other in Angstroms (the one in the text).
Overall, the manuscript has certain merits, since the resulting membrane shows promising results. However, the characterization of the functionalized MWCNTs is questionable and the explanation for the increase flux of water are not convincing. I suggest a major revison of the manuscript before acceptance.
Author Response
Response to Reviewer 2
Reviewer #2:
- The manuscript is reporting the use of a functionalized MWCNT as useful component in a membrane for desalination uses. The authors introduced a long hydrocarbonated chain in order to facilitate the passage of purified water through the membrane. The particularity and originality of the manuscript resides in the particular additive used: 12-chloro-12-oxododecanedioic acid-methyl ester. One aspect that would need some clarification in the Introduction paragraph is why exactly this compound? Why not using a simple fatty acid acyl chloride in the reaction with MWCNT-CH2-OH to obtain a functionalized MWCNT in which the hydrocarbonated chain is linked through an ester function or a fatty amine in the reaction with MWCNT-COOH (in which the hydrocarbonated chain is linked through an amide function)? What would be the expected role of the terminal -COOMe group?
Authors’ reply: We thank the reviewers for this constructive comment. Our group has carried out a series of simulation studies on single-walled and multi-walled carbon nanotubes. The results showed that 100% salt rejection could be achieved and high water flux could be maintained when certain types and number of functional groups (-COOH, -CONH2 and -NH2) were added to the tip and/or interior of single-walled 1.356 nm or 1.763 nm CNTs. While for carbon nanotubes with a diameter greater than 2nm, long-chain groups need to be added to achieve 100% salt rejection. Thus, our research group further implemented experiments by grafting modification (grafting -COOH, -COCl,-CONH2 and -NH2) groups on 0.9 nm and 3.5 nm SWCNTs and 4.9 nm(average diameter) MWCNTs. The experimental results showed that the dispersibility and stabilization of the functionalized CNTs were significantly improved, the water flux and desalination rate of the SWCNTs and MWCNTs doped PA membrane were better than the PA membrane. In particular, the desalination rate of the SWCNTs membranes was higher than that of the equal functional group of the MWCNTs membrane, which might attribute to the larger diameter of MWCNTs and the comparatively short functional group. Therefore, we speculated that the water flux and desalination rate will be increased by grafting long aliphatic chain on MWCNTs. In this paper, we conducted experiments to graft long aliphatic chain (12-chloro-12-oxododecanedioic acid-methyl ester, containing two ester groups, can make water molecules form hydrogen bonds, which is expected to improve the hydrophilicity of MWNTs) on MWCNTs to explore the effects of water flux and desalination rate of membranes.
2.What is the irrefutable indication the functionalization occurred inside the CNT and not on the outer walls, which would be more probable?
Authors’ reply: We thank the reviewers for their construction comments. In this paper, functionalization could occur both inside and entrance of the CNTs. The groups may have probable to occur outside wall of CNT. From the TEM and inner diameter distribution curve of MWCNTs and MWCNT-CH2OCOC12H23O3 (Fig.3), the reduction in the inner diameter of CNTs revealed the presence of modified in inner walls.
- Why the authors assume throughout the text the presence of a 12-oxododecanedioic acid, when the synthesis was stoped at an ester moiety? There are no indication of further hydrolysis. Moreover, the formula used by the authors corresponds to an ester.
Authors’ reply: We thank the reviewers for their construction comments. This is a mistake in our description. The grafting groups of MWCNTs in this paper are 12-chloro-12-oxododecanedioic acid-methyl ester, which is an ester not acid. we did not further hydrolyze to obtained 12-oxododecanedioic acid.
- The IR spectra need a better description. Fig 2b is showing the formation of the monoester acylchloride more clearly through the peaks at 1740 cm-1 and 1803 cm-1 which represent the stretching vibration of carbonyl C=O bond specific to acyl chlorides (1803), respectively ester (1740). The C=O absorption of -COOH group is more toward the 1700s.
Authors’ reply: We thank the reviewers for their construction comments. This is a mistake in the naming of the compounds in Figure 2. The name of the compounds in Figure 2 should be 12-chloro-12-oxododecanedioic acid-methyl ester, which is an ester not an acid. The strong C=O stretching vibration of most saturated fatty acid esters occurs at 1740 cm-1 higher than the normal frequency for ketones.
5.The presence of a single 12-oxododecanedioic acid-methyl ester strain inside the CNT is, in my opinion, not enough to “clog” the CNT and explain the reduced “diameter” (in fact, according to authors’explanation, in would be in fact a “half-diameter”). I suggest another possible explanation for the reduced diameter: imagine that not one, but two such long chained substitutents face each-other on opposite sides of the inner wall; could atractions such hydrogen bonding (there are -COOCH3 moieties at each end) or hydrophobe-hydrophobe interactions between the hydrocarbonated long chains could explain the reduction in diameter?
Authors’ reply: We thank the reviewers for their construction comments. We agree with the reviewer, two such long chained substitutents face each-other on opposite sides of the inner wall; could atractions such hydrogen bonding (there are -COOCH3 moieties at each end) or hydrophobe-hydrophobe interactions between the hydrocarbonated long chains could explain the reduction in diameter. And The simulation results with more chains added are shown in Fig.4b.
6.A problem is that the authors assume that: “Due to addition of the long chain functional groups of 12-oxododecanedioic acid at the inner wall of the CNTs, the hydrophobic CNTs were converted into hydrophilic ones, which made the water molecules more accessible.” I disagree with that statement: first the moiety introduced is not an acid, but an ester. Second, it has a long hydrocarbonated chain, which should increase the hydrophobicity (not the hydrophilicity). Third, the inner diameter of the MWCNTs is smaller, according to the authors’ own measurments. I suggest another possible explanation: the -COOCH3 terminal group allows the admission and passage of water. In the same time, the smaller diameters observed for the modified MWCNT does not represent a shrinked CNT, but shows that inside these there some kind of bottle-neck formations – these could act as some kind of miniature Venturi tubes that accelerates tha passage of water. Since the molecular diameter of a water molecule is about 2.75A (27.5 nm – interestingly close to the 28 nm inner diameter of the modified MWCNTs) there is plenty of room for these to pass through the space between the long-chain esters and the inner walls of the CNT, even within the reduce space in the bottle-necked formations. On the other hand, if the functionalization occurred on the outer walls, could the pressure of a large group, associated with repulsions between the terminal -COOCH3 group and vicinal other CNTs explain the diminished inner diameter?
Authors’ reply: We thank the reviewers for their construction comments. The long chain functional groups of 12-oxododecanedioic acid-methyl ester, containing two ester groups, can make water molecules form hydrogen bonds easier, which is expected to improve the hydrophilicity of MWNTs. To test our hypothesis, we performed 1ns molecular dynamics simulations and statistics hydrogen bond distribution of MWCNT and MWCNT-CH2OCOC12H23O3. Results showed that MWCNT-CH2OCOC12H23O3 shown more hydrogen bond (Fig.S2).
General comments:
-1. The text should be re-edited for proper English language and clearer presentation. I wil give only one example, but the entire manuscript must be corrected.
“20g dodecanedioic acid (99%) was dissolved in 100 mL methanol solution and then 2 mL concentrated sulfuric acid was added. After being treated for 24 h by stirring reflux at 70℃, the solvent was distilled by a rotary evaporator, and the distilled residues were poured into the ice water, then extracted with ethyl acetate for three times and combined with organic layer which was washed with 10% sodium hydrogen carbonate and saturated salt water in turn. In the end, after drying with anhydrous magnesium sulfate and filtering, the filtrate was poured into the rotator evaporator to remove the solvent ethyl acetate. Subsequently, the colorless oily liquid dimethyldodecanedioate was obtained.” - “Dodecanedioic acid (20 g, 99% purity) was dissolved in 100 mL methanol solution after which 2 mL concentrated sulfuric acid were added. After stirring for 24 h at reflux (70℃), the solvent was distilled in a rotary evaporator, and the distillation residue was directly poured into ice water, subsequently extracted with three portions of ethyl acetate. The combined organic layers were first washed with 10% NaHCO3 solution and then with a saturated salt solution. Finally, after drying on anhydrous magnesium sulfate and subsequent filtration, the filtrate was poured into the rotatory evaporator to remove the excess organic solvent. Thus, the brute dimethyldodecanedioate was obtained as a colorless oily liquid.”
Authors’ reply: We thank the reviewers for their construction comments. The manuscript had been re-edited for proper English language and clearer presentation, especially for the methods section.
-2. In the Experimental part, in the description of the syntheses, give also the moles in brackets for each reagent – it will help to establis the molar ratio between the reagents for each step of the process, since the synthesis implies a complete esterification, followed by a statistical mono-hydrolysis (be the way, how the authors can be sure that by-products such as dodecanedioic acid and unreacted dimethyldodecanedioate were also present in the reaction mixture?)
Authors’ reply: We thank the reviewers for their construction comments. We have added the moles in brackets for each reagent. The extra dodecanedioic acid was reaction with potassium hydroxide to form water-soluble substances and removed by extraction. Unreacted dimethyldodecanedioate was removed by filtered and washed with water and ethanol.
-3. there is a discrepency between Figure 1 (the synthesis steps) and the nomenclature used: what exactly is the end point of the synthesis, the acid (as in the text) or the ester (as in Fig.1). In other words, is the moiety in C1 a -COOH or a -COOCH3 group? From the general formula used for the functionalized MWCNT I assume it is the ester, but the authors should clarify this and therefore use the proper names in the text.
Authors’ reply: We thank the reviewers for their construction comments. This is a mistake in the naming of the compounds in the text. We have revised the names in the text.
-4. I assume that the long chain oxododecanedioic acid is grafted on the MWCNTs through a SN2 reaction between the -CH2OH moiett and the acyl chloride. Therefore, I think it would be more suitable, since the starting material was designated as MWCNT-CH2-OH, to write the reaction product as MWCNT-CH2-O-CO-C12H23O2. For example, it would be clearer in a lict such as “MWCNTs, MWCNT-COOH, MWCNT-OH and MWCNT-C14H25O4”, which would become “MWCNTs, MWCNT-COOH, MWCNT-CH2-OH and MWCNT-CH2-OCO-C12H23O3”.
Authors’ reply: We thank the reviewers for their construction comments. We have revised the licts “MWCNTs, MWCNT-COOH, MWCNT-OH and MWCNT- C14H25O4” to “MWCNTs, MWCNT-COOH, MWCNT-CH2OH and MWCNT-CH2OCOC12H23O3”.in the text.
-5. “The product of 12-chloro-12-oxododecanedioic acid was added into the round bottom flask, then 15 mL DMF and 15 mL pyridine were mixed as solvent and acid binding agent, and then 200 mg MWCNT-CH2OH were added. After keeping a continuous reaction for 2 h in ice water, the product MWCNT-C14H25O4 was filtered and then washed with water and ethanol separately and dried at 60℃for 12 h.” – what was the amount of ester used in the functionalization process? What was the amount of fatty ester grafts introduced on the MWCNTs? Eventually this could be determined by knowing the starting quantity of 12-chloro-12-oxododecanedioic acid methyl ester used and by measuring the quantity found in the solution after filtration.
Authors’ reply: We thank the reviewers for their construction comments. The amount of ester used in the functionalization process was 5.4331g. The amount of fatty ester grafts introduced on the MWCNTs was about 47.56%, which was calculation by TGA analysis. From 100°C to 550°C, the -CH2OCOC12H23O3 in MWNT- CH2OCOC12H23O3 decomposes with a weight loss of about 32.23%.
Minor comments:
- separation of “polymer” is not “pol – ymer”, but “po-lymer” or “poly-mer”; same observation for “properties” – “pro-perties” or “proper-ties”; same observation many other strange such separations
- use “mL” as IS symbol for milliliter
- “water layer” – “aqueous layer”
- “stratrificated” ???
- the correct name of the end-product of the synthesis is not “12-chloro-12-oxododecanedioic acid”, but “12-chloro-12-oxododecanedioic acid-methyl ester”
- leave some blank space between figures 3a and 3b; same for fig. 7a-c
- fig 4 presents a length of 20.63 nm for the hydrocarbonated -CH2-O-CO-C10H20-COOCH3 chain, while in the text the value that is given is 2.063 nm; which is correct? I assume one is in nm and the other in Angstroms (the one in the text).
Authors’ reply: We thank the reviewers for the detailed comments.
-We revised the strange separations of words.
- We used “mL” symbol instead of “ml”.
- We changed “water layer” to “aqueous layer”.
- We revised the error word “stratrificated” to “stratified”.
- We changed the name of end-product “12-chloro-12-oxododecanedioic acid” to “12-chloro-12-oxododecanedioic acid-methyl ester”.
- The units of Fig.4 is Å, and we labeled in title of Fig.4.

Round 2
Reviewer 2 Report
Dear Sir,
The authors have operated some modifications (as required), but avoided others.
- The term “12-oxododecanedioic acid” is still extensively used throughout the text, although it is obvious that the grafted moiety is its methyl ester.
- The IR analysis has not been improved by clearly identifying (as suggested) the -COCl, respectively -COOMe moities.
- there were (and still are) some discrepancies in the authors’ assumptions and explanations to which I have disagreed and for which I have proposed the authors alternative explanations. In their answer they have agreed with me, but in the manuscript there were no changes.
- I have asked for a complete review of the English language and gave specifically only one example. Modification was made according to this single example, only in the specified paragraph, but no other correction in the text was operated, the numerous awkward sentencing remaining untouched and uncorrected. I could eventually spent a few days and point out all these strange phrasing and offer correct formulations, but I don’t think that this is the reviewer’s job.
I cannot therefore consider that the manuscript is ready for publication and maintain my opinion for a major revision, centered on the Results and Discussion paragraph.
Author Response
Response to Reviewer 2
Reviewer #2:
The authors have operated some modifications (as required), but avoided others.
- The term “12-oxododecanedioic acid” is still extensively used throughout the text, although it is obvious that the grafted moiety is its methyl ester.
Authors’ reply: We thank the reviewers for this constructive comment. We have changed the name of end-product “12-chloro-12-oxododecanedioic acid” to “12-chloro-12-oxododecanedioic acid-methyl ester” and marked with blue color.
- The IR analysis has not been improved by clearly identifying (as suggested) the -COCl, respectively -COOMe moities.
Authors’ reply: We thank the reviewers for this constructive comment. We have improved description of IR analysis, and clearly identifying the existence peaks of -COCl and -COOMe, respectively.
- there were (and still are) some discrepancies in the authors’ assumptions and explanations to which I have disagreed and for which I have proposed the authors alternative explanations. In their answer they have agreed with me, but in the manuscript there were no changes.
Authors’ reply: We thank the reviewers for this constructive comment. We have revised the relevant content in the manuscript and marked in blue in lines 313 to 323.
- I have asked for a complete review of the English language and gave specifically only one example. Modification was made according to this single example, only in the specified paragraph, but no other correction in the text was operated, the numerous awkward sentencing remaining untouched and uncorrected. I could eventually spent a few days and point out all these strange phrasing and offer correct formulations, but I don’t think that this is the reviewer’s job.
I cannot therefore consider that the manuscript is ready for publication and maintain my opinion for a major revision, centered on the Results and Discussion paragraph.
Authors’ reply: We thank the reviewers for this constructive comment. We have significantly revised the full text, especially, the Results and Discussion paragraph marked with red color.

Round 3
Reviewer 2 Report
Dear Sir,
The manuscript can now be accepted for publication.
Author Response
We greatly appreciated this reviewer for his/her positive comments.